# Importance of Bioactive Substances in Sheep’s Milk in Human Health

**DOI:** 10.3390/ijms22094364

**Published:** 2021-04-22

**Authors:** Zuzanna Flis, Edyta Molik

**Affiliations:** Department of Animal Nutrition and Biotechnology, and Fisheries, Faculty of Animal Science, University of Agriculture in Krakow, 31-059 Krakow, Poland; zuzanna.flis@student.urk.edu.pl

**Keywords:** functional foods, natural bioactive molecules, natural products, sheep, milk

## Abstract

Sheep’s milk is an important source of bioactive substances that have health-promoting functions for the body. The valuable composition of sheep’s milk is due to the high content of fatty acids, immunoglobulins, proteins, hormones, vitamins and minerals. Many biopeptides found in milk have antibacterial, antiviral and anti-inflammatory properties. The bioactive substances of sheep’s milk also show anticancer properties. Sheep’s milk, thanks to its content of CLA and orotic acid, prevents the occurrence of type 2 diabetes, Alzheimer’s disease and cancer. Sheep’s milk, as a product rich in bioactive substances, can be used as a medical aid to support the body in the fight against neurological and cancer diseases.

## 1. Introduction

Nowadays, a sedentary lifestyle, lack of physical activity and an inadequate diet contribute to the development of many diseases. In recent years, consumer awareness of foods that contain biologically active ingredients and directly affect health has increased. Functional food plays a key role in human health [1]. Sheep’s milk and its products are an important source of fatty acids, calcium, phosphorus, iron and magnesium [2,3]. According to Lordan et al. (2018) and Megalemou et al. (2017), fermented milk products have a health-promoting effect, especially on the cardiovascular system and civilization diseases [4,5]. Consuming yogurt and kefir reduces obesity and the risk of metabolic syndrome [6] and prevents type II diabetes [7]. Sheep’s milk also contains many biologically active, antibacterial, immunomodulatory and antioxidant substances. The high content of valuable nutrients and biologically active substances in sheep’s milk prove the dietary value of this milk and products derived from it.

Fermented beverages made from sheep’s milk are characterized by good antibacterial properties, due to the presence of bioactive compounds such as catechol, vanillin, ferulic acid and salicylic acid. Additionally, kefir is rich in vitamins such as B1, B2, B5 and C; minerals; and amino acids that are essential for the healing process and homeostasis [8]. These dairy products can be used as an adjunct therapy in the treatment of neurological tumors (glioma) [9]. A study by dos Reis et al. (2019) on Wistar rats has shown that consumption of kefir reduced the occurrence of aberrant crypt foci by 36% [10]. Thus, the administration of kefir to animals helped to reduce the development of lesions, possibly by increasing short-chain fatty acid production, reducing intestinal permeability and improving colonic antioxidant activity [10]. Besides anticancer properties, fermented drinks (yogurt, kefir) have a positive effect on the wound healing process [11], have antibacterial properties [12] and modulate the intestinal microflora [13]. Recent studies show that kefir consumption also improves exercise capacity and reduces postexercise lactic acid production in humans [14]. Sheep’s milk is used in the production of prebiotics and probiotics and as an ingredient in infant formulas, thus being a suitable alternative to mother’s milk and for the production of nutraceuticals [15]. Owing to the content of bioactive substances, sheep’s milk can be used for the production of medical food.

## 2. Methodology

At the initial stage of preparing this review, similarly to Guiné et al. (2020) and Guiné et al. (2021), the topic to be addressed was selected [16,17]. After searching available literature, it was concluded that there is a need to gather in a review article up-to-date information on the importance of sheep’s milk bioactive substances on human health, which is scattered around the scientific literature. Then, by selecting the appropriate keywords, the following scientific databases were searched: Web of Science, Science Direct, Scopus and PubMed. For each of the read publications, certain principles were established for their inclusion in this review [16,17]. The most up-to-date scientific articles and those that best suited the issues of the review were selected. The final version of the article presents references from 1979 to 2021. Particular emphasis was placed on papers published within the last 5 years, which represent over 50% of the included publications.

## 3. The Role of Milk Proteins in the Functioning of the Body

Milk proteins include casein complexes (80%) and whey proteins (20%). The whey protein fraction contains several extremely important ingredients, including lysozyme, lactoperoxidase or lactoferrin [18]. Research by Caboni et al. (2019) showed that sheep’s milk contains the most lactoferrin and proteins that have antibacterial and anti-inflammatory effects [19]. Whey proteins exhibit immunoactive properties and include α-lactalbumins, β-lactoglobulins, immunoglobulins, lactoperoxidase (LP), lysozyme and lactoferrin (LF) [20]. Lysozyme and α-lactalbumin show reactive oxygen species (ROS)-dependent cytotoxicity in many tumor cells such as MCF-7, MDA-MB231, HeLa and MG 63 [21]. In vitro and in vivo studies by Li et al. (2020) have shown that lactoferrin, α-lactalbumin and β-lactoglobulin exhibit neuroprotective effects by increasing bopindolol levels followed by inhibition of the TLR4-related pathway [22].

LF as a sheep’s milk protein has antibacterial, antioxidant, anticancer and anti-inflammatory effects [23,24]. Research by Zheng et al. (2020) proves that LF has the ability to alleviate oxidative stress in the hippocampus [25]. Studies have shown that administration of LF regenerates cells of the CA1 subregion of the hippocampus in elderly mice. LF has antiviral activity against, among others, rotavirus, HIV and the hepatitis C virus [26,27]. LF has a high affinity for iron ions [28]. According to Sanchez et al. (1992), LF is the most important protein that binds iron, thus blocking the growth of some microorganisms and the development of infections [29]. Additionally, LF has the ability to inhibit the proliferation of cancer cells [18]. LF can bind to transferrin receptors (TFR) and LF membrane internalization receptors (LFR) that are characteristic of cancer cells [30]. The expression of the LF receptor occurs in the capillaries of the brain; additionally, LF has the ability to cross the blood–brain barrier. This suggests that LF is an excellent ligand in the field of nanomedicine that can deliver substances to the brain [31,32]. These properties of LF are used in the design of targeted drug delivery systems that can percolate cancer cells or cross the brain barrier [30,31,33,34]. LF can be used to treat glioblastoma. It has been shown that bovine LF inhibits the migration of human glioblastoma cell lines by reducing SNAIL and vimentin expression, thereby increasing cadherin levels and inhibiting the IL-6/STAT3 axis [35]. The studies of Eliassen et al. (2002) prove that LF has significant cytotoxic activity against the colon cancer line C26, thus confirming the antitumor activity of this protein [36].

An important endogenous amino acid of sheep’s milk protein is proline, which plays a key role in the synthesis of arginine and polyamines and activates mTOR cell signaling to initiate the process of protein synthesis, especially collagen [37]. Both proline and hydroxyproline are found in the highest amounts in sheep’s milk proteins (Table 1) and in collagen. The demand for this amino acid increases in the prenatal period and just after birth, so it is crucial for the proper growth and development of young organisms. Proline and hydroxyproline account for as much as 12% of proteins in the body of a newborn [38]. A study by Singh et al. (2017) showed an indirect effect of proline on the proliferation of MCF-7 and MDA-MB-231 breast cancer cells [39]. Proline plays an important role in the synthesis of polyamines in the placenta of pregnant sheep [37]. Sheep’s milk is characterized by a high content of proline which influences the synthesis of hemoglobin. The proline-rich polypeptide (PRP) participates in the humoral immune response and is responsible for the maturation of regulatory T lymphocytes (Tregs). PRP can also inhibit the synthesis of amyloid-beta (Aβ) and reduce its toxic effects on nerve cells [40]. Aβ, a peptide consisting of 40–42 amino acids, is a major component of amyloid plaques in the brain of patients with Alzheimer’s disease [41]. Aβ molecules, which are formed from amyloid precursor protein (APP), have the ability to self-aggregate, creating various forms of oligomers that are toxic to nerve cells. Some incorrectly folded oligomers cause a chain reaction and change the degree of the folding of subsequent Aβ molecules. The most common type of neurodegenerative disorder is Alzheimer’s disease, which is characterized by cognitive decline and memory impairment. This is due to the deposition of Aβ in neuritic plaques and its further effects on microglia, astrocytes, neurons and the post-translational modification of the Tau protein [42,43]. Research by Bharadwaj et al. (2013) has shown that dairy products rich in proline protect against Alzheimer’s disease and other amyloidogenic diseases [41]. In a rat model of Alzheimer’s disease, there is an increased level of oxidative stress in cells and raised concentration of brain monoamines, which is associated with structural damage to monoaminergic neurons. In animals treated with intramuscular injections of PRP, a partial reversal of neurodegenerative changes was observed [44].

## 4. The Importance of Fatty Acids

For many years, attention has been paid to the importance of fatty acids in preventing the risk of cancer. Fatty acids are one of the most important bioactive components of mammalian milk. Due to their high nutritional value and influence on the physicochemical processes of the body, they are necessary for the proper development of the nervous system and the growth of a young organism [46]. Polar lipids (PLs) are an important component of milk fat with significant pro-health properties. Although the phospholipid fraction of sheep’s milk is quantitatively a minor component of the overall lipid content (Table 2), it is extremely important as it exhibits anticoagulant activity [47]. Research by Megalemou and Sioriki (2017) and Poutzalis et al. (2016) has shown that the most active inhibitors of platelet-activating factor (PAF) were found in yogurt made from sheep’s milk and goat’s milk [5,48]. The processing of milk, especially the fermentation process, enhances the antithrombotic properties of PLs against PAF [47] and thrombin-induced platelet aggregation [49]. The antithrombotic activity of PLs was also demonstrated in a study on the production of traditional Greek sheep cheeses such as Ladotyri and Kefalotyri [50]. Due to their properties, polar lipid fractions improve human health by reducing the level of atherogenic lipoprotein cholesterol, modulating the intestinal microflora and reducing inflammation in blood serum and liver [51]. Studies on low-density lipoprotein (LDL) receptor knockout mice have shown that the addition of milk to feed reduced the development of atherosclerosis compared to animals fed a high-fat diet without milk [51]. Recently, yogurts with the addition of omega-3 fatty acids (O-3FAs) have been gaining popularity. Recent studies show that consuming yogurt with O-3FAs significantly reduces the risk of developing infectious diseases [52]. Sheep’s milk contains short- and medium-chain fatty acids (representing a fraction of about 11%), which are extremely important for a healthy human diet [53,54]. Sheep’s milk has higher concentrations of butyric acid (C4:0), omega-3 fatty acid and conjugated linoleic acid (CLA) than milk from other ruminants [54].

A special role in the regeneration of the nervous system is attributed to conjugated linoleic acid dienes [55]. It is believed that the cis-9, trans-11 octadecadienoic acid (rumenic acid) is the major isomer of CLA present in the milk fat of ruminants (Table 3). Its amount can be up to 90% of total CLA [1,55]. Of all the ruminants, sheep’s milk turns out to be the richest (1.1%) in CLA, but the concentration of this component in milk depends on the season [56,57]. According to Zervas et al. (2011), sheep’s milk has a higher CLA content than goat’s milk, due to differences in the mRNA of the adipocytes of the mammary glands in both species [58]. In recent years, evidence has been provided largely based on in vitro studies and human clinical studies that CLA, besides its classical mechanism of action mediated by nuclear transcription factors, also exhibits a number of interdependent molecular signaling pathways that are responsible for human health [59]. CLA inhibits both benign and malignant tumors by inhibiting cell growth and development [60]. According to reports by Ochoa et al. (2004), the 10-CLA isomer acts peripherally by modulating the apoptosis process and controlling the cell cycle, while the major 9-CLA isomer influences the metabolism of arachidonic acid [61]. Cellular mechanisms modulating CLA synthesis may modify cell proliferation, lipid oxidation and vitamin A transformations [62,63,64]. It has been shown that women who consume high-fat dairy products have a lower incidence of colon cancer, and this may be partly related to the high CLA content in these foods [65]. Administration of LF-CLA complex to rats with Alzheimer’s disease resulted in a 2-fold decrease in Aβ in the hippocampus. This effect may be due to the ability of CLA to destroy existing Aβ and inhibit the formation of new oligomers of this protein. In addition, in the group treated with the LF–CLA complex, there was a significant decrease in the levels of reactive oxygen species (ROS), nitrite (NO) and malondialdehyde (MDA), which indicates the antioxidant effect of CLA and the ability to scavenge free radicals. All the obtained results of oxidative stress markers showed a high antioxidant capacity of the LF–CLA complex due to its active ability to cross the blood–brain barrier (BBB) and target the brain tissue. By reducing the level of arachidonic acid (ARA), CLA can influence the growth of cancer cells. For this reason, rats treated with the LF–CLA complex showed significantly lower levels of TNF-α [30]. Similar results regarding the effect of CLA on oxidative stress, MDA levels and ARA levels have also been observed in other studies [66,67]. The mother’s diet during pregnancy and lactation can have a significant impact on the brain development of the offspring. CLA is delivered to the fetus through the placenta during pregnancy and through breast milk in the infancy period [68]. Administration of CLA to rats during pregnancy and lactation caused a reduction in brain lipid peroxidation in the offspring [67].

Some bioactive substances contained in sheep’s milk have a health-promoting effect on the body due to their self-healing properties. Genomic and mitochondrial DNA molecules are continually exposed to damage from endogenous metabolites, environmental carcinogens, some anti-inflammatory drugs and genotoxic cancer therapeutics [70]. This damage can cause permanent changes to the DNA molecule that preclude the cell from transcribing the damaged DNA fragment. When DNA repair processes are not working efficiently, the generation of DNA lesions and mutations leads to carcinogenic transformation [71]. The main sources of endogenous DNA damage are ROS and alkyl groups [72]. However, there are mechanisms that correct damaged DNA molecules. The DNA damage response involves the activation of complex signaling networks that repair DNA damage and maintain genome integrity [70,73]. This is crucial in preventing tumorigenesis [73]. A study by Izzotti et al. (2003) on newborn mice showed that the sudden transition from maternal-mediated respiration to autonomic pulmonary respiration in the fetus causes a significant increase in extensive DNA adducts and oxidative DNA lesions in the lungs [74]. This DNA damage was attenuated by the upregulation of many genes involved in oxidative stress and DNA repair. Additionally, prenatal administration of the antioxidant N-acetylcysteine prevented all transcriptional changes in the lungs [74], suggesting links between oxidative stress, DNA damage and tissue function. It is widely accepted that an impaired DNA damage repair system is a common mechanism in neurodegenerative diseases [75]. A large accumulation of DNA damage occurs most often in the central nervous system due to the low DNA repair capacity in postmitotic brain tissue [76]. Accumulation of DNA damage is a well-known factor of aging and, therefore, is believed to be the main cause of Alzheimer’s disease [77]. Orotic acid is an important precursor in the biosynthetic pathway of pyrimidine nucleotides and, therefore, participates in the synthesis of DNA and RNA [78]. In mammals, it is released from mitochondrial dihydroorotate dehydrogenase (DHODH) by the enzyme cytoplasmic synthase UMP for conversion to UMP [79]. Cows’ milk contains the most orotic acid, followed by sheep’s milk and then goat’s milk. In addition, the content of this component in milk depends on the season of the year, the stage of lactation and the breed of the animal [80,81]. In ruminants, the highest concentration of orotic acid in milk is observed in the middle of lactation [78]. Still, little is known about the function of orotic acid in milk and its impact on human health; it can be assumed that it is needed in developing the microbiome in the stomach of ruminants [78]. It has been shown that orotic acid improves learning ability in adult rats [82], and it has a neuroprotective effect in gerbils and cats with transient cerebral ischemia [83]. Additionally, the orotic acid molecule is thought to be needed to regulate genes that are extremely important in the development of cells, tissues and organisms [79].

## 5. The Importance of Vitamins and Hormones

Sheep’s milk contains a high content of B vitamins (Table 4), which are important in the development of the brain and the functioning of the nervous system [54,56,84,85]. The latest research by Ryan et al. (2020) shows that patients with depression have lower concentrations of vitamins B3 and B6 and high levels of IL-6 and CRP proteins [86]. Vitamin B5 (D-pantothenic acid) is a precursor to coenzyme A (CoA), which plays a key role in biological processes that regulate the metabolism of carbohydrates, lipids, proteins and nucleic acids. In the brain, acetyl-CoA is relevant for the synthesis of myelin and the neurotransmitter acetylcholine [87]. Vitamin B12 deficiency during pregnancy can cause adverse effects on the child’s development, such as speech disorders and concentration problems [88]. Vitamin B12 regulates glial migration and synaptic formation by controlling the expression specific for the PTP-3/LAR PRTP (leukocyte common antigen-related receptor-type tyrosine-protein phosphatase) isoforms. It has also been found that the absorption of vitamin B12 provided through the diet is crucial for the expression of the long PTP-3 (PTP-3A) isoform in neuronal and glial cells. PTP-3A expression autonomously regulates glial migration and synaptic formation through interaction with the extracellular matrix protein NID-1/nidogen 1 [89].

The key hormones found in milk and influencing the proper development of offspring include leptin, ghrelin, insulin growth factor 1 (IGF-1), adiponectin and insulin [90]. Insulin, leptin and adiponectin are mainly responsible for regulating the energy balance. According to Schneider-Worthington et al. (2020), levels of these hormones may be related to maternal body mass index [91]. This is confirmed by the research of Kugananthan et al. (2017), which showed a correlation between maternal obesity and higher levels of leptin and protein in milk [92]. This may be due to the secretion of more leptin into the bloodstream by adipose tissue and thus an increase in the concentration of leptin in the milk. Similarly, in malnourished mothers, low levels of this hormone in milk are observed [93], which significantly affects the development of the nervous system of the offspring [92]. Leptin receptors present in the hippocampus can affect learning and memory [94]. In addition, leptin is also considered to play a neurotrophic role because its deficiency has severely disrupted the development of the arcuate nucleus of the hypothalamus [95]. It was found that leptin potentiates the neurogenic process in the hippocampus and the subventricular zone of the brain in a mouse study on Alzheimer’s disease. Leptin treatment increases neural stem cell (NSC) proliferation and leads to attenuation of Aβ-induced neurodegeneration and the production of superoxide anions. Researchers have emphasized the need for further research to elucidate which signaling mechanisms are involved in the neurogenic and neuroprotective effects of this hormone, which could lead to the development of new therapies for the treatment of Alzheimer’s disease in the future [96]. Sahin et al. (2020) reported that leptin induces synaptogenesis in the developing hippocampus through increased expression of KLF4 and cytokine signaling suppressor 3 (SOCS3) in hippocampal neurons [97]. Additionally, it exhibits neuroprotective effects and alleviates spatial memory impairment in rat studies of premature brain injury, which may contribute to a better understanding of the protective effect of this hormone in premature infants with cerebral hypoxia [98,99].

In the case of adiponectin, no relationship was found between maternal obesity and the concentration of this hormone in milk [92]. This suggests that adiponectin is mostly synthesized in the mammary gland [100]. It is considered that some bioactive hormones found in milk, including adiponectin, link the metabolic status of the mother with the metabolic health of the offspring [101]. Milk-fed infants had higher levels of adiponectin and had less weight gain in the first 6 months of life [102]. It is believed that adiponectin influences the proper course of fatty acid metabolism and indicates anti-inflammatory properties [103]. Adiponectin may also have neuroprotective effects on oxidative stress-induced brain damage [104]. It is suggested that adiponectin may mediate AdipoR1/APPL1/LKB1/AMPK signaling, because the administration of recombinant human exogenous adiponectin (rh-adiponectin) to young mice after cerebral infarction reduces neuronal apoptosis [105].

Insulin plays an important role in regulating the distribution of nutrients in ruminants during lactation [106]. Plasma insulin levels also affect the milk yield of ruminants; namely, low insulin levels are correlated with higher milk production [107]. Li et al. (2020) proved in their study that administering donkey milk powder to rats with type 2 diabetes for 4 weeks significantly increases target organ insulin sensitivity, lowers blood glucose levels, improves insulin resistance, increases the ability to capture free radicals and improves the level of antioxidants in the body [22]. Wingrove et al. (2019) reported that inhaled insulin administration uses the nose–brain pathway and delivers the drug directly to the brain tissue while limiting systemic exposure [108]. Insulin administered in this way appears to improve the potential of the brain’s mitochondrial membrane and stimulates the activity of the brain’s mitochondrial complexes in a streptozotocin-induced model of early type 2 diabetes [109]. Injections of lipopolysaccharide (LPS) into the brain ventricles of rats caused inflammation of the nervous system. The cognitive functions were impaired and the levels of IL-1β and TNF-α increased in the cortex and hippocampus of the animals. Studies have shown that LPS modulates mitochondrial function and induces oxidative stress by reducing the activity of superoxide dismutase and catalase and the levels of glutathione and sulfhydryl. Treatment with insulin caused a reversal of all effects, which proves the potential role of insulin as a therapeutic drug in inflammatory diseases related to mitochondrial dysfunction in the brain [110].

## 6. Importance of Milk MicroRNA

Micro-ribonucleic acids (miRNAs) are small non-coding RNA molecules 18–25 nucleotides long, responsible for regulating 40% to 60% of gene expression at the post-transcriptional level [111]. The miRNA precursor is transcribed primarily by RNA polymerase II and then is processed to mature miRNAs [112]. miRNAs are responsible for regulating protein expression by binding to complementary mRNAs and then directing the mRNA to degrade or inhibit translation [113]. Studies have shown that miRNA-21 (a mammalian microRNA encoded by the MIR21 gene) can control the proliferation and apoptosis of many cell types, including cancer cells [114]. As demonstrated by Rasoolnezhad et al. (2021), miR-128-5p inhibits proliferation in breast cancer cells by regulating the PI3K/AKT pathway [115]. Moreover, miRNA-138-5p induced apoptosis by increasing the levels of caspase-9 and caspase-3 and stopping the cell cycle in the sub-G1 phase. It has been found that miRNA-21 can act on glioblastoma cells. This is due to the inhibition of caspase-3 and caspase-7 expression in vitro and in vivo. miRNA plays an important role in the growth and development of animals and performs a variety of biological functions in their tissues and organs [116]. miRNAs were detected in saliva [117], urine [118] and milk [119]. The synthesis of miRNAs takes place in the mammary gland; then, when the cubs drink milk, the miRNAs are transported to the intestine, where they are absorbed by epithelial cells [120]. miRNA molecules enter tissues and organs through the circulatory system so that they can perform various functions, e.g., immunoprotection [121]. The miRNA in milk is in the form of free molecules packed into extracellular vesicles (EVs) such as exosomes that protect against degradation [121,122,123]. According to Zhou et al. (2012), more than 60% of all pre-miRNAs that are associated with resistance are found in breast milk [119]. miRNA plays a key role in the physiology of the mammary gland and in the lactation of livestock [31,124], but its function in sheep is not fully understood. In sheep, the miRNA is presumed to be responsible for the growth of the fleece [125] and muscle [126]. A study by Hao et al. (2021) on Gansu Alpine Merino and Small-Tailed Han sheep showed that differently expressed miRNA target genes were largely involved in some metabolic and signaling pathways related to mammary gland development and synthesis of milk proteins and fats [127]. Previous studies on the effects of miRNAs on the mammary gland focused primarily on identifying specific miRNAs as oncogenes and tumor suppressors that regulate gene expression by targeting mRNAs in breast cancer [128]. Human milk is a rich source of miRNAs that are specific to lactation, and this is why they began to be used as biomarkers of mammary gland efficiency [121]. Genc et al. (2018) examined the changes in the expression of miRNAs in subacute sclerosing panencephalitis (SSPE) [129]. IL-29 and miR-548 levels were increased in SSPE patients. The increased expression of miR-548 may be a compensatory result of an excessive immune system response. This proves that IL-29 and miR548 can be involved in the pathogenesis of this disease and can be used in the diagnosis and treatment of SSPE. Oral administration of milk to mice causes detectable accumulation of EVs in their tissues, mainly the liver and brain [130]. It has been found that miRNA-9-5p is an important regulator of angiogenesis following traumatic brain injury (TBI). Studies in rats have shown significantly higher levels of miRNA-9-5p and an increased density of vessels and neurons in the damaged areas. Elevated expression of miRNA-9-5p promoted angiogenesis in the injured cerebral cortex and restored neurological functions by activating the Hedgehog pathway and increasing expression of p-AKT [131].

## 7. Summary

Sheep’s milk is a product rich in bioactive substances needed for the proper development of young organisms. The valuable composition of sheep’s milk is due to the high content of fatty acids, immunoglobulins, proteins, hormones, vitamins and minerals. Due to having the highest linoleic acid content of all ruminants, sheep’s milk is effective in preventing obesity, type 2 diabetes and cancer. Many active biopeptides found in milk have proven antiviral, antibacterial and anti-inflammatory properties. In addition, they also exhibit cytotoxic activity against cancer cells. A specific feature of sheep’s milk is the high level of vitamins, especially B vitamins and minerals. The effect of sheep’s milk on the human body is still not fully understood, and the latest research focuses on the importance of orotic acid and microRNAs that are present in the milk of these ruminants. Undoubtedly, sheep’s milk is a rich source of health-promoting substances for the human body, and thanks to this, it can act as an effective functional food.

## Figures and Tables

**Table 1 ijms-22-04364-t001:** Amino acid composition of sheep’s milk proteins (adapted from [45]).

Amino Acids	In g/100 g of Sheep Milk	In g/100 g of Casein
Tryptophan	0.084	1.3
Threonine	0.268	3.6
Isoleucine	0.338	5.1
Leucine	0.587	9.0
Lysine	0.513	7.3
Methionine	0.155	2.1
Cysteine	0.035	0.8
Phenylalanine	0.284	5.2
Tyrosine	0.281	5.6
Valine	0.448	6.7
Arginine	0.198	3.3
Histidine	0.167	3.3
Alanine	0.269	3.2
Aspartic acid	0.328	7.7
Glutamic acid	1.019	21.1
Glycine	0.041	1.7
Proline	-	10
Serine	0.492	5.0

**Table 2 ijms-22-04364-t002:** Content of total lipids (TLs), total polar lipids (TPLs) and total neutral lipids (TNLs) expressed in grams per 100 g of sheep milk and yogurt (adapted from [47]).

Dairy Products	TLs (g/100 g)	TNLs (%TL)	TPLs (%TL)
Sheep Milk	5.28 ± 0.37	95.15 ± 2.30	3.20 ± 0.56
Yoghurt A	8.10 ± 0.43	96.46 ± 1.07	2.45 ± 0.20
Yoghurt B	8.23 ± 1.59	97.62 ± 0.22	2.29 ± 0.17
Yoghurt C	7.23 ± 0.60	97.47 ± 0.53	2.10 ± 0.37
Yoghurt D	7.47 ± 0.36	97.34 ± 0.47	2.25 ± 0.10
Yoghurt E	9.20 ± 0.55	97.60 ± 0.38	2.55 ± 0.45

**Table 3 ijms-22-04364-t003:** Conjugated linoleic acid (CLA) isomers (% total CLA) in sheep’s milk (adapted from [69]).

Isomer	Sheep’s Milk
trans-12, trans-14	1.31–3.47
trans-11, trans-13	1.21–5.08
trans-10, trans-12	1.17–1.77
trans-9, trans-11	1.13–1.99
trans-8, trans-10	1.05–1.37
trans-7, trans-9	0.48–0.61
12–14 (cis–trans plus trans–cis)	0.52–1.83
11–13 (cis–trans plus trans–cis)	0.76–4.23
10–12(cis–trans plus trans–cis)	0.28–0.41
9–11 (cis–trans plus trans–cis)	76.5–82.4
8–10 (cis–trans plus trans–cis)	0.11–0.71
7–9 (cis–trans plus trans–cis)	3.31–9.69

**Table 4 ijms-22-04364-t004:** Average content of vitamins in 100 g of sheep’s, cow’s and goat’s milk (adapted from [45]).

Vitamin	Sheep’s Milk	Cow’s Milk	Goat’s Milk
Retinol (µg)	64 ± 19.5	35.0 ± 8.0	0.04 ± 0.0
Carotenoids (µg)	Trace amounts	16.0 ± 8.0	Trace amounts
Vitamin A (µg)	64.0 ± 5.5	37.0 ± 8.0	54.32 ± 0.00
Vitamin E (mg)	0.11 ± 0.01	0.08 ± 0.01	0.04 ± 0.0
Thiamin (mg)	0.07 ± 0.01	0.04 ± 0.01	0.059 ± 0.0
Riboflavin (mg)	0.3 ± 0.02	0.2 ± 0.01	0.175 ± 0.0
Niacin (mg)	0.41 ± 0.05	0.13 ± 0.05	0.235 ± 0.0
Pantothenic acid (mg)	0.43 ± 0.02	0.43 ± 0.12	0.31 ± 0.0
Vitamin B6 (mg)	0.07 ± 0.01	0.04 ± 0.01	0.048 ± 0.0
Folic acid (µg)	6.0 ± 0.06	8.5 ± 1.5	1.0 ± 0.0
Biotin (µg)	2.5 ± 0.0	2.0 ± 0.5	1.75 ± 0.3
Vitamin B12 (µg)	0.66 ± 0.05	0.5 ± 0.3	0.065 ± 0.0
Vitamin C (mg)	4.6 ± 0.4	1.0 ± 0.5	1.295 ± 0.0
Vitamin D (µg)	0.2 ± 0.0	0.2 ± 0.1	0.15 ± 0.1

## Data Availability

Not applicable.

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
