# Peer review of "Importance of Bioactive Substances in Sheep’s Milk in Human Health"

_ijms, 2021, doi:10.3390/ijms22094364_

Round 1

Reviewer 1 Report

Relevance of the review and organization of the topics:

This review is interesting and highlights the scientific evidence supporting the benefits for the human health, and particularly for brain health, of the bioactive compounds present in sheep’s milk. On the other hand, the organization according to the type of components is beneficial to better present the information and facilitate reading and interpretation.

Title, Abstract and keywords:

The abstract does not correctly fit the main core and objective of the review, as it is mentioned in the title. It is very brief and when it comes to evidencing the health benefits of sheep milk it mentions effects related to diabetes, or cancer, and only very briefly mentions the objective of this study, which is the effect in brain health. Therefore, the abstract has to be improved and some of the most important findings about the effect of the different components of sheep milk in brain health, not in other diseases.

The number of keywords is adequate in number, but in a n article about brain development, the keyword cancer is questionable because it is not the core of the article.

From these initial elements, it seems that the title does not match the abstract and does not match the keywords. It is very confusing, because we do not get any idea of a coherent work coming from this.

Please consider revising and modifying the title, see my comments below.

Introduction:

The introductory section has adequate information to introduce the topic of the review. However, it is very limited. What is the point of having an introductory section constituted by one single paragraph? It clearly must be improved and extended further to justify being a separate section, which must be present no doubt.

Methodology:

Although being a review, this should include a methodology section explaining how the review was conducted. You can do the review following systematic review methodology or not, but even in this last case you have to explain how you got to this point. Please read and cite the following articles to help you introduce a methodology section:

  • Guiné RPF, Barroca MJ, Coldea TE, Bartkiene E, Anjos O (2021) Apple Fermented Products: An Overview of Technology, Properties and Health Effects. Processes, 9(2), 223: 1-27.
  • Guiné RPF, Florença SG, Barroca MJ, Anjos O (2021) The duality of innovation and food development versus purely traditional foods. Trends in Food Science & Technology, 109(1), 16-24.
  • Guiné RPF, Florença SG, Barroca MJ, Anjos O (2020) The Link between the Consumer and the Innovations in Food Product Development. Foods, 9(9), 1317:1-22.

Besides, from this list of works, two are from articles published in MDPI journals, which is beneficial to make a favourable decision in later acceptance of your work.

Development:

As previously referred, the organization of the information according to different types of compounds is positive. However, again it is found that the article is very much focused in general health effects, like antioxidant, anti-inflammatory or antibacterial. Also it is given a great emphasis to cancer again, or virus infections. Therefore, I wonder if the title should be changed in accordance to the contents, and make it more vast and not so focused on brain health, but human health: Try for example: “Importance of bioactive substances in sheep's milk in human health”

Summary

Again the summary does not highlight any special brain development effects of the milk components, so you really must use a different title.

Level of English:

The use of English is correct.

References

The work is adequately referenced, but some additional references were suggested to improve the work, as described above.

Author Response

The given line numbers relate to the "Track Changes" function

  • As suggested, the title was changed from "Importance of bioactive substances in sheep's milk in brain development" to "Importance of bioactive substances in sheep's milk in human health" - verse 2
  • Removal of the word "cancer" from the keywords - line 16
  • To expand Introduction, sentences have been added to the first paragraph - lines 24-27
  • As suggested, a new paragraph has been added to elaborate on the Introduction - verse 31-44
  • The missing methodology section was added, the suggested literature was used - lines 48-59. The papers added are in References under numbers 16 and 17 - line 404 and 406
  • The addition of a new point "Methodology" changed the numbering of the remaining headings in lines 60, 121, 222, 297, 343
  • Zotero programe has automatically updated the numbering of the new references

Reviewer 2 Report

This is a review on sheep's milk and its impact on brain development.

The authors have produced a well-written manuscript.

Some points for improvement are:

  1. There is no information on the lipidomics of sheep milk. An overview table on the polar and neutral lipids should be included.
  2. Another point of improvement, would be to include also the fermented sheep milk products, e.g. yoghurt, kefir etc.

Suggested references to be included

  1. https://www.sciencedirect.com/science/article/abs/pii/S1756464619300295
  2. https://www.ncbi.nlm.nih.gov/pmc/articles/PMC7071183/

I suggest major revision and I am happy to review the revised version.

Author Response

The given line numbers relate to the "Track Changes" function

  • As suggested, a new paragraph has been added with information on fermented sheep's milk products (yoghurt, kefir).
  • As suggested, an overview table on the polar and neutral lipids has been added - verse 122
  • A paragraph on lipidomics of sheep milk has been added. The proposed literature was used (References under numbers 8 and 47 - line 381 and 480) and a few new papers were added - line 128-144
  • Adding a new table changed the numbering of the remaining tables in lines 149 and 223
  • Zotero programe has automatically updated the numbering of the new references

Round 2

Reviewer 1 Report

I have checked that the authors have made modifications in their manuscript according to my previous recommendations and they have significantly improved the work. Therefore my present recommendation is to accept for publication. 

Reviewer 2 Report

the MS is now revised and the authors have taken into consideration all the points raised.

The MS can now be accepted.